# Patient-Relevant Costs for Organ Preservation versus Radical Resection in Locally Advanced Rectal Cancer

**DOI:** 10.3390/cancers16071281

**Published:** 2024-03-26

**Authors:** Georg W. Wurschi, Alexander Rühle, Justus Domschikowski, Maike Trommer, Simone Ferdinandus, Jan-Niklas Becker, Simon Boeke, Mathias Sonnhoff, Christoph A. Fink, Lukas Käsmann, Melanie Schneider, Elodie Bockelmann, David Krug, Nils H. Nicolay, Alexander Fabian, Klaus Pietschmann

**Affiliations:** 1Department of Radiotherapy and Radiation Oncology, Jena University Hospital, 07747 Jena, Germany; klaus.pietschmann@med.uni-jena.de; 2Clinician Scientist Program, Interdisciplinary Center for Clinical Research (IZKF), Jena University Hospital, 07747 Jena, Germany; 3Cancer Center Central Germany (CCCG), 07747 Jena, Germany; 4Department of Radiation Oncology, University of Freiburg—Medical Center, 79106 Freiburg, Germany; 5Department of Radiation Oncology, University of Leipzig Medical Center, 04103 Leipzig, Germany; 6Cancer Center Central Germany (CCCG), 04103 Leipzig, Germany; 7Department of Radiation Oncology, University Hospital Schleswig-Holstein, 24105 Kiel, Germanyalexander.fabian@uksh.de (A.F.); 8Department of Radiation Oncology, Cyberknife and Radiotherapy, Faculty of Medicine and University Hospital Cologne, 50937 Cologne, Germany; 9Center for Molecular Medicine Cologne, University of Cologne, 50931 Cologne, Germany; 10Center of Integrated Oncology, Universities of Aachen, Bonn, Cologne and Düsseldorf (CIO ABCD), 50937 Cologne, Germany; 11Department of Radiotherapy, Hannover Medical School, 30625 Hannover, Germany; 12Department of Radiation Oncology, University Hospital Tübingen, 72076 Tübingen, Germany; 13Center for Radiotherapy and Radiation Oncology, 28239 Bremen, Germany; 14Department of Radiation Oncology, University Hospital Heidelberg, 69120 Heidelberg, Germany; 15Department of Radiation Oncology, University Hospital, LMU Munich, 81377 Munich, Germany; 16Comprehensive Pneumology Center Munich (CPC-M), German Center for Lung Research (DZL), 81377 Munich, Germany; 17German Cancer Consortium (DKTK), Partner Site Munich, 81377 Munich, Germany; 18Department of Radiotherapy and Radiation Oncology, Faculty of Medicine and University Hospital Carl Gustav Carus, TUD Dresden University of Technology, 01307 Dresden, Germany; 19Department of Radiotherapy and Radiation Oncology, University Hospital Hamburg-Eppendorf, 20251 Hamburg, Germany

**Keywords:** rectal cancer, watch and wait, total neoadjuvant therapy, cost-effectiveness, financial toxicity

## Abstract

**Simple Summary:**

Patients with locally advanced rectal cancer and complete remission after “total neoadjuvant therapy” may undergo a “watch and wait” schedule instead of radical resection, but need to be followed-up more frequently. This is the first work analyzing patient-related costs in this setup in the German health care system compared to standard chemoradiotherapy and resection. In this model, patients undergoing watch and wait had a better quality of life, but experienced additional costs from more frequent follow-up visits. Overall, these were cumulatively less than individual costs for medication and ostomy care after radical resection. Thus, organ preservation appeared to be efficacious and cost-effective from a patient’s point of view in the German health care system.

**Abstract:**

Total neoadjuvant therapy (TNT) is an evolving treatment schedule for locally advanced rectal cancer (LARC), allowing for organ preservation in a relevant number of patients in the case of complete response. Patients who undergo this so-called “watch and wait” approach are likely to benefit regarding their quality of life (QoL), especially if definitive ostomy could be avoided. In this work, we performed the first cost-effectiveness analysis from the patient perspective to compare costs for TNT with radical resection after neoadjuvant chemoradiation (CRT) in the German health care system. Individual costs for patients insured with a statutory health insurance were calculated with a Markov microsimulation. A subgroup analysis from the prospective “FinTox” trial was used to calibrate the model’s parameters. We found that TNT was less expensive (−1540 EUR) and simultaneously resulted in a better QoL (+0.64 QALYs) during treatment and 5-year follow-up. The average cost for patients under TNT was 4711 EUR per year, which was equivalent to 3.2% of the net household income. CRT followed by resection resulted in higher overall costs for ostomy care, medication and greater loss of earnings. Overall, TNT appeared to be more efficacious and cost-effective from a patient’s point of view in the German health care system.

## 1. Introduction

Resection has been the backbone for rectal cancer for years. Advances in the treatment of locally advanced rectal cancer by multimodal treatment, i.e., neoadjuvant or adjuvant use of radiotherapy (RT), chemotherapy (CTx) or combining both (chemoradiotherapy, CRT), lead to remarkable local control rates and a good oncological outcome [1]. Since a couple of years, organ preservation schedules were increasingly recognized for patients obtaining complete remission (CR) after neoadjuvant treatment. For these, “watch and wait” approaches were proven to be safe [2,3]. These organ preservation approaches postpone surgery to the event of local recurrence. The long-term local recurrence rate is approximately 25% for patients obtaining CR [3]. Organ preservation approaches are thus likely to offer a benefit regarding quality of life (QoL) to a remarkable number of patients, especially if sphincter-sparing surgery is impossible due to distal tumor location (<4–6 cm from anal verge) [4,5,6]. Patients thus tended to prefer non-operative management (NOM) when asked to decide between CRT + resection and NOM [7]. These decisions can, of course, not completely be assigned to the patients themselves. Nevertheless, the best-possible information about treatment characteristics and impact on daily life should be provided to the patients in order to improve decision quality and minimize decisional regret [8]. Patient-relevant costs may differ considerably, as this relatively new approach currently requires more frequent follow-up visits in order to timely detect local regrowth [9]. Patients undergoing resection are not required to undergo these examinations so often [1]. There are several cost-effectiveness analyses consistently showing an economical benefit from a third-party view [10,11,12,13,14]. At this point, there are no analyses available from the patient’s point of view. Financial toxicity is generally not routinely monitored or considered for treatment decisions, although patients are likely to experience financial toxicity due to a loss of income or additional expenses for medication or health care products [15,16]. As rectal cancer treatment is mostly intended to be curative, a patient’s long-term financial toxicity has to be considered. Depending on health insurance policies, treatment and follow-up costs may be covered by health insurances, but there may be significant differences between several countries. Many industrialized countries offer social security programs, often including statutory health insurances (SHI, German “Gesetzliche Krankenversicherung”). This provides access to expensive treatments, like multimodal treatment of rectal cancer, for inhabitants irrespective their income. For example, most German citizens are insured with these SHIs [17]. In this system, SHI are reimbursed by income-related premiums. Routine treatment is usually covered by the German SHIs. Nevertheless, not all expenses of cancer treatment are covered by these SHIs. For example, co-payments for medication or health care measures are usually required, driving expenses for consultations might not be refunded in any case or loss of income due to absence from work is limited in time. Additional expenses were thus often seen to occur during cancer treatment in a German multicenter study, although the resulting financial toxicity was rarely not compensable by the patients [15,16,18].

We sought to address this topic by performing a cost-effectiveness analysis from the patient’s point of view, assuming a typical setting of a patient insured with a German SHI. This work should contribute to better informing patients about the possible benefits and upcoming expenses in the shared decision-making process.

## 2. Materials and Methods

We performed a cost-effectiveness analysis for patients with locally advanced rectal cancer after neoadjuvant treatment with a five-year follow-up. Two standard base cases were simulated (see below) and the outcome of these cases was simulated with a Markov simulation. The simulation was computed with RationalWill 6.1.10 (SpiceLogic Inc., Toronto, ON, Canada), SPSS Statistics 29 (IBM, Armonk, NY, USA) and Excel 2016 (Microsoft, Redmond, WA, USA). This analysis was conducted in accordance with the CHEERS reporting standards [19].

The data for base case modeling were derived from the published literature and German epidemiologic registries, notably the Deutsches Statistisches Bundesamt [17]. General population data were used to estimate mortality from natural causes as well as to estimate the loss of income due to treatment. A subgroup of rectal cancer patients from the young DEGRO (German Society of Radiation Oncology) working party multicenter trial “FinTox” [16,20] and its precursor trial [15] were used to control the epidemiologic assumptions from the German general population (see Appendix A for the composition of the reference cohort).

### 2.1. Base Case Description

The standard patients for this model had newly diagnosed stage III tumor (T3/T4, N+, CRM+), located in the lower third of the rectum (<6 cm from the anal verge). These patients underwent neoadjuvant treatment, which was either consisting of TNT or standard CRT. We assumed they achieved CR after TNT and would opt for NOM. In this setting, NOM was assumed to allow for sphincter saving. APR was performed for resection due to distal tumor location and these patients received a definitive ostomy. All patients underwent adjuvant CT after standard CRT and resection. The probability of CR was high in long-term TNT protocols [21,22] and we thus decided for such a protocol in order to account for the intended organ preservation approach. We simulated a treatment similar to the OPRA trial [22] protocol, which consisted of long-term neoadjuvant CRT with concomitant 5-fluorouracil (5-FU) or Capecitabine, followed by eight cycles consolidation CTx (FOLFOX or CapOX). For the CRT patients, we chose the protocol of the experimental arm of the ARO-04 trial [23], as the overall treatment duration was comparable. These patients also underwent long-term CRT (concomitant 5-FU/Oxaliplatin, Ox), followed by resection and eight cycles adjuvant CTx (5-FU/Ox). The average duration of in-patient treatment for tumor resection (16 nights) as well as for lung or liver metastasectomy (5 nights) was derived from the German Diagnosis-Related Groups (G-DRG) system [24]. We assumed ambulatory treatment for CRT and CTx in both cases, except for supportive in-patient treatment in the case of °III/IV toxicity and in-patient treatment for resection. A follow-up period of five years was simulated.

### 2.2. Treatment Costs in the German Health Care System

We assumed that the sample patients were insured with a German SHI, as the majority (2021: 89% [25]) of the German general population is insured with SHIs and the benefits of SHIs are regulated by law and thus relatively homogenous. Self-employed persons or those, earning above a certain threshold, may opt for private insurances in Germany. We did not incorporate these private insurances into this model, as insurance benefits and premiums are rather inhomogeneous and may significantly differ from SHI, depending on the individual contracts.

Patient-related costs for uncovered treatment expenses, driving expenses, costs for health care measures, uncovered medication expensed or loss of salary were estimated. As income and loss of salary depend on the employment status, we calculated the cost for patients who are still employed as well as for patients who were retired and are thus not at risk of being incapable of working. Furthermore, a “mixed model” was calculated with respect to the fraction of retired and working patients based on the age-dependent incidence. The social security system ensures a continued pay of salary for the first six weeks of absence from work without loss of salary [26] and a sick pay of 70% of gross income for another 72 weeks [27]. We assumed the patients returned to work after resection—or in the case of NOM, after a typical period—as well as a temporary disability for local recurrence or resectable distant recurrence. In the case of local and distant recurrence or unsalvageable recurrence, we assumed a permanent disability due to frequent treatments and loss of general health status.

A co-payment for treatment expenses, such as in-patient hospital stays, medication or health care measures, such as ostomy care, of 10 EUR per unit is usually required from patients within the German SHIs [28]. A maximum of 280 EUR is charged for in-patient treatment per year. There is a co-payment exemption limit of 2% of annual gross household income, which can be lowered to 1% annually in the case of chronic diseases [29]. Cumulative costs for co-payments were calculated up to this limit, assuming 2% for primary treatment and 1% for the following years, especially in the case of metastatic or recurrent disease. For patients with local and distant recurrence, we assumed continuous therapy with frequent co-payment needs and set the costs at the co-payment exemption limit. Beyond that, personal costs, which were not covered by insurances, occurred from additional expenses related to treatment: We assumed 10 EUR per month per person for additional ostomy care products or individual (non-prescription) medication.

Another relevant fraction of costs resulted from driving expenses, which are only covered when long-term therapies, such as CRT or CTx, are required. As treatment of rectal cancer is only available in tertiary centers in Germany, an average driving distance of 26.3 km was required for the treatment [30]. We assumed the patients utilize taxis or ambulance services for drives, where co-payments of 5 EUR were required for the first and the last drive of a treatment series, respectively. Driving for follow-up visits is usually not recompensed. Driving expenses were then calculated according to the general rate of 0.2 EUR per km [31].

### 2.3. Model Parameters

We constructed two Markov cycles, modelling “watch and wait” patients after TNT (I, “NOM”) and patients after standard CRT plus resection (II, “resection”). Figure 1 illustrates the corresponding Markov simulation models of these transition states. The following transition states were defined:A.Stable disease,B.Stable disease after successful salvage for local recurrence,C.Local recurrence,D.Distant recurrence,E.Local and distant recurrence, andF.Death.

The patients were cycling between these cycles according to the distinct probabilities for each transition state for a total of five cycles (cycle length = 1 year), corresponding to five years of follow-up. 100,000 cases were simulated for each base case. For each state, a distinct effectiveness and patient-related costs was applied. All costs were reported in Euro [EUR]. The effectiveness was measured in quality-adjusted life years (QALYs), which ranged from 0 (death) to 1 (perfect health) and depended on the patient’s burden from ongoing treatment, persisting side effects or persistent measures, such as an ostomy. Utilities and costs were discounted by 3.5% annually according to current recommendations [32].

We derived the transition probabilities and utilities from the original publications of the ARO-04 trial and the OPRA trial as well as from further published peer-reviewed sources or databases, notably from the International Watch and Wait Database (IWWD) [2,3]. The underlying parameters of this model were summarized in Table 1 and Table 2. Patient-related costs and total QoL were calculated from the cumulative transition between the states for patients in a “watch and wait” schedule (I, “NOM”) as well as for patients, who underwent resection after CRT (II, “resection”). A detailed cost report for every single state in both groups (“NOM”, I and “resection”, II) can be found in the Appendix A.

The incremental cost-effectiveness ratio (*ICER* was calculated according to Equation (1), with *C_NOM_* being the costs of patients in a watch and wait schedule and the corresponding effectiveness *E_NO_*_M_ [QALY] as well as *C_RESECTION_* being the costs for patients who underwent resection and the corresponding effectiveness *E_RESECTION_*, respectively [44].
(1)$$ICER=\frac{\left(C_{NOM}-C_{RESECTION}\right)}{\left(E_{NOM}-E_{RESECTION}\right)}$$

A willingness-to-pay (*WTP*) threshold is required to further assess the cost-effectiveness of these treatments. *WTP* reflects the amount of money that patients would be willing to spend for their treatment to obtain one additional QALY. We assumed 6000 EUR per year for five years being the *WTP* threshold for a curative treatment of rectal cancer in German patients, based on surveys on *WTP* for other tumor entities and general health care expenses in Germany [45,46]. As shown by Equation (2), the net monetary benefit (NMB) is the difference between occurring treatment costs (*C*) and the costs the patient would be expected to pay for the obtained QALYs (*E*) based on the *WTP*, [44]. The *NMB* was computed for both treatment options and an incremental *NMB* (*iNMB*) was used for comparison.
(2)NMB=WTP×E(QALY)−C

A positive *NMB* thus indicated a beneficial treatment for patients and the *NMB* was used in this analysis to simultaneously compare both treatment options regarding their costs and effectiveness (QALY).

## 3. Results

### 3.1. Epidemiology of Rectal Cancer and Income Statistics

A total of 17,895 cases of rectal cancer were diagnosed in Germany in 2019 [47], with a distinct distribution in the age groups as summarized with Table 3. Overall, patients were more often male (11,062 cases, 62%). Patients older than 65 years (64.9%) were considered retired. The rate of persons employed in this group was derived from German general statistics [17] and was then used to calculate the probability of patients in employment with rectal cancer (28.2%) being at risk for a loss of income by the treatment. The mean gross income per age group is furthermore reported in Table 3 and was used to compute an age-dependent income, which was corrected for the age-dependent incidence (Table 4). A mean gross income of 4409 EUR was calculated. The average pension for patients older than 65 years was 1048 EUR (2022, [17]). Having set 80% of the pension as taxable according to the current taxation of pensions, this income was below the German tax allowance per person (2022: 10,347 EUR [48]). No further correction for taxing was therefore required. We calculated an average net household income for this model considering the age-dependent income or pension at 2580 EUR per month (Table 4).

Co-payment in German SHIs is dependent on the household income, including a personal allowance of 5922 EUR per person (mean household size = 2 persons [17]). We calculated a co-payment exemption limit of 411 EUR/month for chronic diseases (in the case of repeated treatments for recurrent disease) and 822 EUR/month as a general threshold (applied for all other states).

### 3.2. Analysis of Patients with Rectal Cancer from the “FinTox” Trial

We analyzed *N* = 44 patients from the “FinTox” trial [16] and its precursor trial [15]. Table 5 summarizes the characteristics of this group. The median age of patients was 67 years (58–75 years interquartile range, IQR) and 50% of these were older than 65 years. Most patients (38/44, 86%) were insured with a German SHI and the following results were reported for this subgroup. The median net household income in this group ranged between 1701 and 2600 EUR (Figure 2) and 10/35 patients (29%) reported a loss of income due to cancer treatment. Additional expenses related to cancer treatment affected 16/32 patients (50%) and mostly ranged between 100 and 500 EUR. The main causes for additional expenses during treatment were driving expenses (17 answers). Of note, 24/35 patients (69%) insured with a SHI did not experience financial difficulties due to cancer treatment.

### 3.3. Cost-Effectiveness Analysis

Total treatment costs were computed for both treatment options (I, TNT and NOM, vs. II, CRT and resection) for all patients as well as for a subgroup of patients, who were in employment, and patients, who were already retired (Table 6). Treatment costs were higher for patients in the resection group (II) in all cases (1017 EUR–1985 EUR), whereas the utility of NOM (I) was rated higher (+0.64 QALYs) for the whole follow-up period of five years, indicating a better overall QoL. Patients, who were in employment, experienced the highest costs (15519 EUR for NOM, 17,504 EUR for resection), as the loss of income was only relevant for those who were not already retired. The *NMB* was thus substantially lower for patients who were in employment than for patients who were retired. Of note, both treatment options yielded a positive *NMB*, meaning the treatment was cost-effective for all patients regardless of the treatment option and employment status.

The transition probabilities of this microsimulation were illustrated by Figure 3 as well as Appendix A. For both treatment options, the calculated overall mortality was in accordance with the underlying studies (NOM 15%, resection 20%) [22,23,34,35,36]. As a result of the model’s simplified design, a transition from certain states of recurrence to stable disease after successful salvage was possible as well as a transition to death during following the cycles. Thus, the cumulative probabilities for states other than death were not comparable with the underlying studies. In both treatment options (I, II), most of the overall costs resulted from the stable states (Appendix A). Within this state, driving expenses were the relevant item except for the working subgroup for NOM (I), where loss of income had the greatest impact. For the resection group, additional expenses resulted from ostomy care, medication and longer overall absence from work (Appendix A). Consequently, overall treatment costs were higher for (II). Due to the lower frequency of follow-up visits compared to NOM (I), driving expenses were lower for (II). Beyond that, the loss of income due to absence of work was the costliest item in the working subgroup for both treatment options. This item was less relevant in the base model, where all patients were included, because of the higher prevalence in patients older than the typical working population (>65 years, Table 3).

### 3.4. The Sensitivity of the Model

Modifications to the model’s parameters were made for the NOM (I) and the resection (II) groups (Table 7). NOM (I) remained dominant over resection (II) regarding the NMB as well as costs and QoL for a broad range of assumed values. As depicted by Figure 4a, NOM (I) is both cheaper and more cost-effective than resection (II). If (I) was costlier than (II), (I) would still remain more cost-effective as long as the costs were below 10,092 EUR. If (I) was even more expensive, (II) would dominate (I). Contrarily, if (II) was less costly than (I), (I) would dominate until (II) was cheaper than 871 EUR (Figure 4b). For the subgroup of retired patients, (II) would never dominate over (I) by reducing costs.

The utility of both treatments was varied in a second, more theoretical approach. If the cumulative utility of NOM was rated less than 2.97 QALYs (−0.9 QALYs), resection (II) would have become more cost-effective. Contrarily, (II) would have dominated (I) in terms of a much better QoL (+0.9 = 4.13 QALYs). These findings were similar in the working and the retired subgroup, respectively.

## 4. Discussion

To the best of our knowledge, this work is the first cost-effectiveness analysis from a patient’s point of view for patients with LARC undergoing NOM after TNT compared with standard CRT and resection. In our opinion, financial toxicity of patients is highly relevant but not adequately examined for most standard treatments. Particularly in the case of rectal cancer treatment, shared decision making after TNT would be extremely welcome to further involve individual patients. Information about patient-relevant costs is useful to provide the best-possible information and enhance the decision-making basis, which is currently dominated by clinical outcome data. Patient acceptance of organ preservation appears to be good and the majority of patients would favor NOM over resection in the case of CR, as reported earlier [5,7]. Nevertheless, individual patient’s costs from absence from work, driving expenses or additional medication have never been reported quantitatively.

Assessing a patient’s individual financial toxicity in such a theoretical approach is difficult due to the heterogeneity of the individual patient’s situation and regional differences in health care systems. Earlier cost-effectiveness analyses for NOM were thus reporting costs from a third-party’s view [10,11,12,13,14,49,50]. In the German health care system, financial toxicity was found to be the result of either treatment costs that were not covered by the health insurances (such as costs for co-payments, additional medication or driving expenses) or the loss of income [16]. The insurance benefits of German SHI’s are standardized by law. We thus limited the model to patients insured with a SHI. The results of this model cannot be easily applied for patients in other countries as health insurance is not standardized internationally. Furthermore, our assumptions for the base cases were based on current general population data. This approach allowed for more representative results, but was prone to be biased by differences in the simulated rectal cancer population from socioeconomic factors. To limit this bias, a subgroup analysis of the “FinTox” trial [15,16,18,51] was performed and enabled a comparison with a multicenter real-world rectal cancer population in Germany. Epidemiologic results and income data of this group were in a similar order. The IQRs and intervals for median income included the calculated parameters from general population data and, consequently, did not require us to reject the hypothesis that general population data supplied realistic data for a simulated rectal cancer cohort in Germany. Further testing for differences was not performed due to the small sample size of this cohort.

We found that both treatment options are cost-effective for the underlying WTP, but NOM (I) was more cost-effective in each simulated case. The simulated effectiveness was comparable with values reported in the literature [10,11,12,13,14,49]. Differences occurred from various model’s specifications, as we applied specific utilities and transition probabilities from the ARO-04 trial [35,42] for the base case. NOM was dominant over resection in concordance with other cost-effectiveness simulations [10,11,12,13,14,49]. The model appeared to be insensitive to changes in treatment costs or QALYs in one-way exploratory sensitivity modeling, suggesting that NOM (I) remained the preferred treatment option regarding the patient’s costs and QoL even with varied parameters.

The higher number of follow-up visits increased costs for these patients, whereas ostomy care costs were avoided. Avoiding definitive ostomy reduced long-term costs for patients in our model as well as third-party costs, such as for health insurances [52,53]. Carlsson et al. reported long-term ostomy costs of 166,700 SEK per year for the health care system (equivalent to approx. 14,884 EUR, 11.2 SEK: 1 EUR exchange rate) and costs for patients of 9100 SEK (813 EUR) per year for early retirement or 37,400 SEK (3339 EUR) per year for sickness absence [52]. In sum, NOM was less costly despite more frequent follow-up visits and the resulting higher driving expenses or longer absence from work.

This model is limited by the assumptions for the base cases. We focused on distal tumors (<6 cm from the anal verge) and assumed a non-sphincter-sparing resection, as the benefit for NOM was more obvious. This model may hence not be fully appropriate for more proximal tumors, requiring temporary ostomy after resection only. Regarding the QoL, NOM yielded slightly higher QALYs per year (0.8) than total mesorectal excision (TME) with temporary ostomy (0.75) or abdominoperineal resection (APR) with definitive ostomy (0.7) [10,43]. The overall benefit of NOM over resection may hence be slightly smaller in patients with more proximal tumors and colostomy reversal.

Overall, rectal cancer treatment costs seem to be substantially covered by German SHIs. Third-party costs for NOM or resection were found to be much higher [11] than the costs reported in our model. As per the specifications of German SHIs, co-payments for medication or ostomy care as well as certain driving expenses were not covered and thus resulted in the reported treatment costs. The costs for NOM corresponded to 3.2% of the net household income for the whole follow-up period, but more than threefold higher costs resulted in the subgroup of working patients. These findings, in particular, need to be interpreted with caution when comparing them with other health care systems, especially in countries, where health insurance is not mandatory or provided for the entire population. In these countries, patients are at risk for substantially higher financial toxicity if treatment costs are not largely covered by the insurance [46].

Treatment or medication costs were widely covered by German SHIs, and overall costs in our model mostly depended on the patient’s employment status. Based on current incidence date, the majority of patients (65%) were older than 65 years and thus not considered as still working. These patients experienced lower costs, as their income from their pension was not affected by cancer treatment. However, a rising incidence in younger men and women has been observed [54,55,56,57,58,59,60], making loss of income likely a more frequent problem in rectal cancer patients in the future.

Taken together, this work demonstrated that cost-effectiveness analysis from a patient’s perspective is realizable with data from published peer-reviewed sources as well as data from the general population. The obtained information may aid participatory decision making for patient-oriented treatment planning.

## 5. Conclusions

Treatment costs are largely covered in the German health care system and did not directly affect patients. Patient-relevant costs mostly resulted from the loss of income and driving expenses for follow-up visits for NOM. Additional costs for stoma care and medication occurred in the resection group, resulting in higher costs in this group. NOM, thus, reduced costs and increased QoL for patients compared with resection after neoadjuvant treatment.

## Figures and Tables

**Figure 1 cancers-16-01281-f001:**
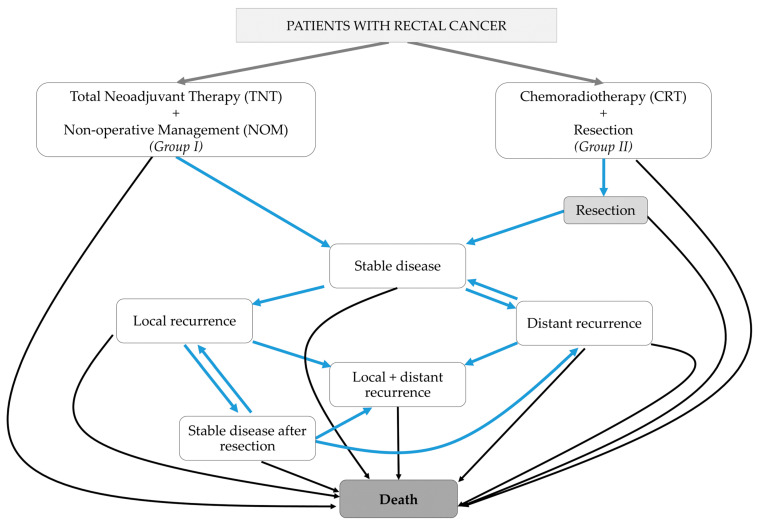
A transition state diagram of the Markov simulation for both treatment options (non-operative management, I and chemoradiotherapy + resection, II). A transition to other states as indicated (blue arrows) or remaining in the current state was possible. Death was referred to as “absorbing state”, as no transition from death to other states was possible (black arrows). CRT—chemoradiotherapy, NOM—non-operative management, and TNT—total neoadjuvant therapy.

**Figure 2 cancers-16-01281-f002:**
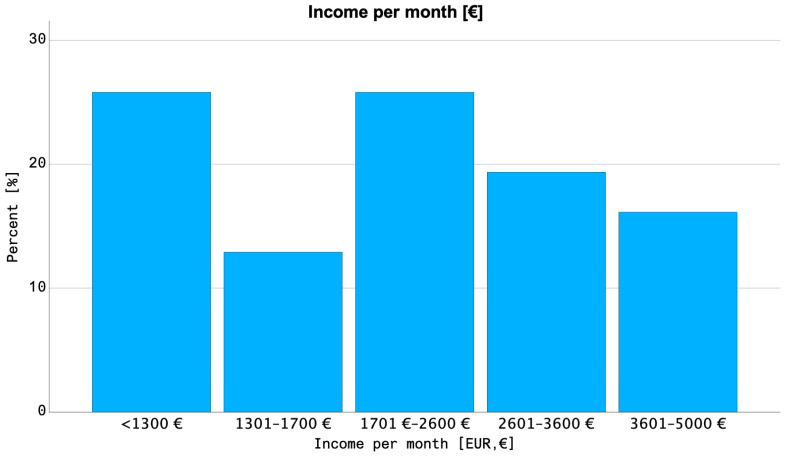
Net household income per month of rectal cancer patients insured with a German statutory health insurance (N = 38) from the reference cohort by Fabian et al. [15,18]. The mean income was found to be 1701–2600 EUR.

**Figure 3 cancers-16-01281-f003:**
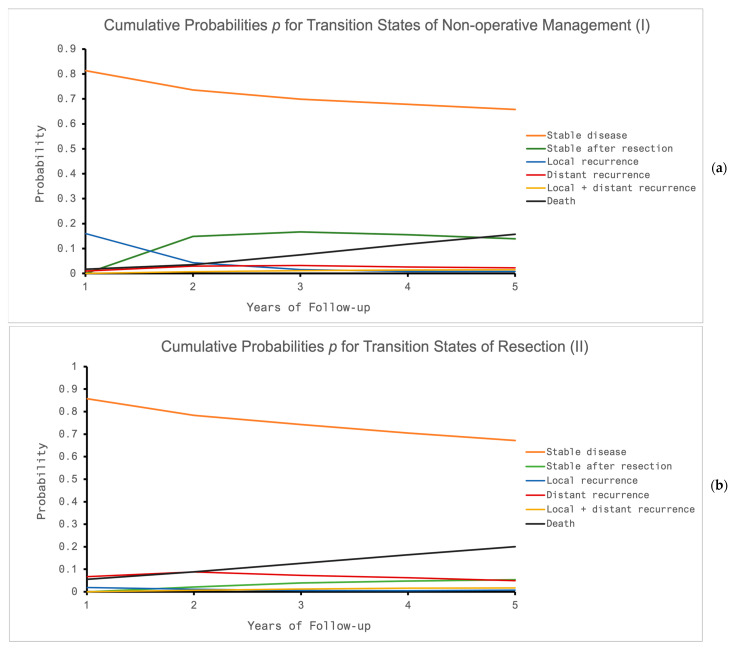
(**a**,**b**) Cumulative probabilities *p* for distinct transition states in the case of (**a**) non-operative management (NOM, I) and (**b**) resection after chemoradiotherapy, CRT, (II) for a follow-up period of five years.

**Figure 4 cancers-16-01281-f004:**
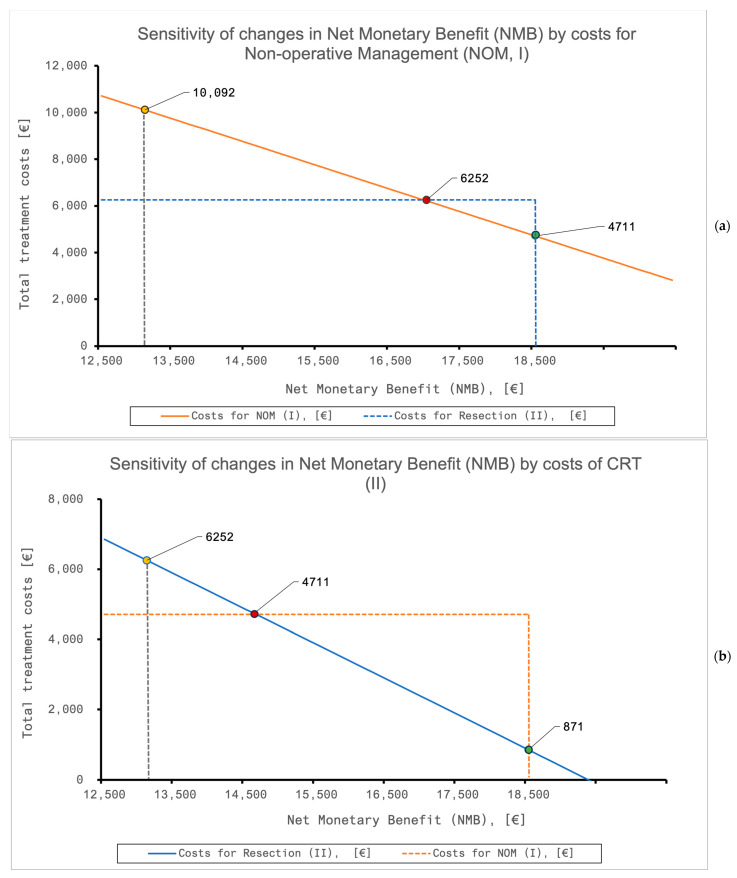
(**a**,**b**) The sensitivity of changes in the costs to the net monetary benefit (NMB) of NOM (**a**) and CRT + resection (II) for all included patients. (**a**) Rising costs of NOM decrease the NMB, so that resection (II, blue dashed line) becomes more cost-effective as soon as NOM (I) cost is >10,092 EUR. (**b**) Resection (II) becomes more cost-effective by reducing costs and would dominate NOM (I, orange dashed line) if costs are <871 EUR. At equal costs for NOM (I) and resection (II), NOM remains dominating. CRT—chemoradiotherapy, NMB—net monetary benefit, and NOM—non-operative management.

**Table 1 cancers-16-01281-t001:** Transition probability parameters used in the models for NOM (I) and resection (II) as well as their respective citation. * Resection for local recurrence. CRT—chemoradiotherapy, NOM—non-operative management, and TNT—total neoadjuvant therapy.

Parameters: (Time Frame)		NOM after TNT (I)		Resection after CRT (II)
	Value	Citation	Value	Citation
Overall grade 3 or 4 toxicity	0.32	Garcia-Aguilar et al. 2022 [22]	0.23	Rödel et al. 2012 [23]
Perioperative death	0.04 *	Marijnen et al. 2002 [33]	0.04	Marijnen et al. 2002 [33]
Local recurrence				
(2 years)	0.21	Van der Valk et al. 2018 [3]	-	
(3 years)	-		0.03	Rödel et al. 2015 [34]
(5 years)	0.24	Van der Valk et al. 2018 [3]	0.05	Diefenhardt et al. 2023 [35]
Distant recurrence		Van der Valk et al. 2018 [3]		
(3 years)	0.04	Van der Valk et al. 2018 [3],	0.19	Rödel et al. 2015 [34]
(5 years)	0.08	Verheij et al. 2023 [36]	0.21	Rödel et al. 2015 [34]
Overall Survival				
(3 years)	0.95	Van der Valk et al. 2018 [3]	0.89	Rödel et al. 2015 [34]
(5 years)	0.85	Van der Valk et al. 2018 [3]	0.79	Rödel et al. 2015 [34]
Distant recurrence when local recurrence occurred	0.18	Van der Valk et al. 2018 [3]	0.17	Diefenhardt et al. 2023 [35]
Local recurrence when distant recurrence occurred	0.54	Van der Valk et al. 2018 [3]	0.36	Diefenhardt et al. 2023 [35]
Local recurrence after salvage resection				
(5 years)	0.13	Ikoma et al. 2017 [37]	0.13	Ikoma et al. 2017 [37]
Distant recurrence after local recurrence				
(3 years)	0.11	Van der Valk et al. 2018 [3]	0.11	Van der Valk et al. 2018 [3]
Salvage for local recurrence	0.94	Dossa et al. 2019 [38]	0.59	Ikoma et al. 2017 [37]
Salvage for distant recurrence	0.57	Ikoma et al. 2017 [37]	0.57	Ikoma et al. 2017 [37]
Mortality after local recurrence and salvage				
(5 years)	0.5	Rao et al. 2017 [39]	0.5	Rao et al. 2017 [39]
Mortality after local recurrence without salvage				
(5 years)	0.7	Rao et al. 2017 [39]	0.7	Rao et al. 2017 [39]
Mortality after distant recurrence				
(5 years)	0.8	Rao et al. 2017 [39]	0.8	Rao et al. 2017 [39]
Mortality after combined Local + distant recurrence				
(5 years)	0.8	Rao et al. 2017 [39]	0.8	Rao et al. 2017 [39]
Mortality of other causes				
(per year)	0.02	DESTATIS Zensus 2022 [17,40]	0.02	DESTATIS Zensus 2022 [17,40]

**Table 2 cancers-16-01281-t002:** Health state utility parameters [QALYs] and their citation. QALY—quality-adjusted life year, ranging from 0 to 1. CRT—chemoradiotherapy, NOM—non-operative management, and TNT—total neoadjuvant therapy.

Utility Parameters:		NOM after TNT (I)		Resection after CRT (II)
	Value	Citation	Value	Citation
Initial state: NOM/surgery after CRT	0.80	Couwenberg et al. 2018 [41], Cui et al. 2022 [10]	0.61	Kosmala et al. 2021 [42,43]
Long term: NOM (no recurrence)/surgery after CRT (no recurrence)	0.80	Couwenberg et al. 2018 [41]	0.70	Couwenberg et al. 2018 [41], Kosmala et al. 2021 [42]
Salvage surgery (local recurrence)	0.70	Couwenberg et al. 2018 [41]	0.70	Couwenberg et al. 2018 [41]
Local recurrence	0.67	Van den Brink et al. 2004 [43]	0.67	Van den Brink et al. 2004 [43]
Distant recurrence	0.70	Van den Brink et al. 2004 [43]	0.70	Van den Brink et al. 2004 [43]
Local + distant recurrence	0.48	Van den Brink et al. 2004 [43]	0.48	Van den Brink et al. 2004 [43]
Death	0	-	0	-

**Table 3 cancers-16-01281-t003:** Number of newly diagnosed rectal cancer cases in Germany (2019, [47]) as well as employment rates and gross income per month in the German general population (2022) per age group. Percentages may not total 100 due to rounding differences.

Age Group	<25	25–<45	45–<65	≥65	Total
Rectal cancer cases in 2019 [47]	14	460	5807	11,614	17,895
Percentage of incidence 2019 [%]	<0.5	2.6	32.5	64.9	100
Employment rate 2022 [%] [17]	38.2	78.0	73.7	3.4	28.2
Gross income per month 2022 [mean, EUR] [17]	2913	3881	4155	4557	4409

**Table 4 cancers-16-01281-t004:** Income and co-payment values derived from general population data and their corresponding reference. Income values for the rectal cancer cohort were calculated from the age-dependent incidence rates and the respective income in these groups as in Table 3).

Income and Co-Payment Values Derived from General Population Data	
Net household income per month 2021 [average, EUR] [17]	3813 EUR
Gross pension payment per month 2022 [average, EUR] [17]	1048 EUR
Reduced earning capacity pension 2022 [average, EUR] [17]	925 EUR
Gross income (age-corrected) for employed patients in the rectal cancer cohort [47], 2022 [mean, EUR]	4409 EUR
Net household income for rectal cancer cohort [47], 2022 [mean, EUR]	2580 EUR
Co-payment exemption limit (mean household size: 2 persons [17])	
1% of gross income (for chronic diseases)	411 EUR/month
2% of gross income (general threshold)	822 EUR/month

**Table 5 cancers-16-01281-t005:** Results of the reference cohort derived from the “FinTox” trial and its precursor trial by Fabian et al. [15,16,18]. The number of patients [N] and corresponding percentage [%] are described for each subgroup. Percentages may not total 100 due to rounding differences. IQR—interquartile range and SHI—statutory health insurance.

Patients with Rectal Cancer [N]		44
Age [years]		
Median		67
Interquartile range (IQR), 25–75%		58–75
Gender [N], (%)		
Female		19 (43%)
Male		25 (57%)
Patients insured with a German statutory health insurance (SHI) [N], (%)		38 (86%)
Employment status of patients insured with SHI	Employed	12 (32%)
	Not employed	4 (11%)
	Retired	18 (47%)
	Not reported	4 (11%)
Net household income of patients insured with SHI	<1300 EUR	8 (21%)
	1300–1700 EUR	4 (11%)
	1701–2600 EUR	8 (21%)
	2601–3600 EUR	6 (16%)
	3601–5000 EUR	5 (13%)
	>5000 EUR	0
	Not reported	7 (18%)
Loss of income related to cancer treatment of patients insured with SHI	None	25 (66%)
	<100 EUR/month	2 (5%)
	100–500 EUR/month	4 (11%)
	500–1500 EUR/month	4 (11%)
	Not reported	3 (8%)
Additional expenses related to cancer treatment of patients insured with SHI	None	16 (42%)
	<100 EUR/month	3 (8%)
	100–500 EUR/month	12 (32%)
	500–1500 EUR/month	1 (3%)
	Not reported	6 (16%)
*Of those:* Reasons for additional expenses *(multiple mentioning was possible)*	Expenses for co-payments	17
	Driving expenses	10
	Expenses for medication or health care products	10
	Other (not specified)	1
Financial difficulties due to treatment in patients insured with SHI	None	24 (63%)
	Little	5 (13%)
	Moderate	4 (11%)
	Great	2 (5%)
	Not reported	3 (8%)

**Table 6 cancers-16-01281-t006:** Results of the cost-effectiveness analysis for treatment option I (total neoadjuvant therapy, TNT, followed by non-operative management, NOM) versus treatment option II (chemoradiotherapy, CRT, followed by resection), applying an annual willingness-to-pay (*WTP*) threshold of 6000 EUR. Costs, quality of life (quality-adjusted life years, QALY) and the net monetary benefit (*NMB*) are calculated for both treatment options for three scenarios (patients currently employed, retired patients and a “mixed model”, computing the average income from the typical employment and retirement rate for this cohort). Both options are compared by the difference in costs (Δ *costs*), the incremental cost-effectiveness ratio (*ICER*) and the incremental (i) *NMB*.

Treatment Option	QALY	Costs	[EUR]	Δ Costs [EUR]	ICER	NMB [EUR]	iNMB [EUR]
TNT + NOM (I)	3.87	All patients	4711	−1540	−2407	18,509	5380
		Employed	15,519	−1985	−1590	7701	5825
		Retired	813	−1017	−3101	22,407	4857
CRT + Resection (II)	3.23	All patients	6252			13,128	
		Employed	17,504			1876	
		Retired	1830			17,550	

**Table 7 cancers-16-01281-t007:** The sensitivity of the model to changes in treatment costs and utility (cumulative 5-year quality-adjusted life years, QALY) to the resulting net monetary benefit (*NMB*) per treatment option. The maximum possible QALY is set at 5 for the follow-up period. The minimal possible cost is 0 EUR. An equivalent *NMB* indicates a comparable effectiveness for both treatment options. * Out of a possible range, NOM remains dominating. NOM—non-operative management.

		QALY	QALY, Where NMB Becomes Equivalent	Costs	Costs, Where NMB Becomes Equivalent
NOM (I)	All patients	3.87	2.97	4711	10,092
	Employed		2.90	15,519	21,344
	Retired		3.06	813	5670
Resection (II)	All patients	3.23	4.13	6252	871
	Employed		4.20	17,504	11,679
	Retired		4.04	1830	<0 *

## Data Availability

Publicly available datasets were analyzed in this study. These data can be found by the corresponding reference. The original contributions presented in this study are included in this article/Appendix A, and further inquiries can be directed to the corresponding author/s.

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
