# Peer review of "Patient-Relevant Costs for Organ Preservation versus Radical Resection in Locally Advanced Rectal Cancer"

_cancers, 2024, doi:10.3390/cancers16071281_

Round 1

Reviewer 1 Report

Comments and Suggestions for Authors

This manuscript essentially describes cost-effectiveness comparison between non-operative management and resection in locally advanced rectal cancer.

The manuscript is valuable in evaluating cost effectiveness and QoL from patient point of view.

As methodology, the estimation was done via Markov model approach. All discussion was based on estimation and no inferential statistics was used. The estimation was strengthened by presenting sensitivity by changes in important parameter, net monetary benefit.

Author Response

Thank you very much for taking the time to review this manuscript and your recommendation to accept our manuscript!

Reviewer 2 Report

Comments and Suggestions for Authors

This study is well-designed and the manuscript is well-written. Thus, I just have two minor suggestions.

1. Please remove "How much is “watch and wait”, doctor? –" from title.

2. Please add the note following each table to show the meaning of all abbreviations.

Author Response

Comments 1:  Please remove "How much is “watch and wait”, doctor? –" from title.

Response 1: The title was corrected as requested and the graphical abstract was modified accordingly.

Comments 2: Please add the note following each table to show the meaning of all abbreviations.

Response 2: We agree and we have, accordingly, revised all table and figure captions in order to include the meanings to all abbreviations.

Reviewer 3 Report

Comments and Suggestions for Authors

This article is a very impressive, well-stuctured, well-designes, well-presented study. The authors compared CRT+surgical treatmet with "watch-and-wait" proticol in the case of rectal cancer patients. This is the first work analyzing patient-related costs in this setup in the German health care system compared to standard chemoradiotherapy and resection. They found that "watch-and-wait" schema reduced costs and increased quality of life for patients compared with resection after neoadjuvant treatment. 

The study is important, well-presented, and of high clinical and financial merit. 

One minor question:

Are patients informed in detail about the advantages and disadvantages of the two types of procedural protocols? How much say does the patient have in the medical decision? 

After minor revision, I suggest to accept the manuscript for publication. 

Author Response

We agree with this comment. In our opinion, implementing a shared decision making process would be welcomed in order to obtain the best-possible results including the patient’s perspective. It is, nevertheless, correct that treatment indication and decision for watch and wait cannot completely be assigned to the patients and more comprehensive information would be need being provided to the patients. The introduction was written ambiguous in that regard. We have revised the introduction and the corresponding discussion accordingly to present our personal view more clearly.
